# Fintech Adoption Factors: A Study on an Educated Romanian Population

Lucian Belascu [1], Corina Anca Negut [2,*], Zeno Dinca [2], Cosmin Alin Botoroga [2] and Dan Gabriel Dumitrescu [2]

1 Department of Management, Marketing and Business Administration, Lucian Blaga University of Sibiu, 550024 Sibiu , Romania; lucian.belascu@ulbsibiu.ro
2 Department of International Business and Economics, Bucharest University of Economic Studies, 010374 București, Romania; dincazeno16@stud.ase.ro (Z.D.); botorogacosmin17@stud.ase.ro (C.A.B.); dan.dumitrescu@rei.ase.ro (D.G.D.)
* Correspondence: negutcorina20@stud.ase.ro

**Abstract:** Even though the literature implies that customers and banking organizations can profit from digital banking in various ways, client adoption of this service is still low, especially in emerging and developing nations. Consumers' openness to digital services limits their willingness to adopt digital banking, especially mobile banking services. We used a quantitative research method based on a questionnaire sent during August–December 2022 to Romanian individuals and received 118 answers, which we analyzed using the logistic regression model; throughout, we determined the extent of mobile banking use, payments, and banking products needed within the population with tertiary education, as well as new developments that the shift to digitalization brings to users, with new features for existing products, cryptocurrency accounts, and fintech companies now being complementary to traditional banks. Our study presents current customer perceptions of implementing bank digitalization through mobile applications in a developing nation like Romania; here, advantages are counterbalanced by limitations and there are, undoubtedly, difficulties to be overcome in the quest for a more effective e-business framework. We determined the factors that are relevant in making people use fintech accounts using logit analysis.

**Keywords:** fintech; bank digitalization; digital services; financial services; electronic wallet; cash; internet banking; cryptocurrencies; virtual cards; mobile banking





## 1. Introduction

With recent developments in technology, the COVID-19 pandemic, and the broad utilization of smartphones, we observe an increased interest among banks in upgrading their infrastructures in order to move part of their products and services online (to digitalize). At the same time, financial products sold by fintech companies are gaining more popularity around the world.

According to Leong 2018 [1], we define fintech as a cross-disciplinary subject that combines finance, technology management, and innovation management. Moreover, in order to discuss how fintech creates value for businesses, we summarized various fintech applications into four major categories: (i) payment, (ii) advisory service, (iii) financing, and (iv) compliance. The definition can further be elaborated as "any innovative ideas that improve financial service processes by proposing technology solutions according to different business situations, while the ideas could also lead to new business models or even new businesses". As a subtopic within payment, blockchain has widely been studied, and many relevant techniques and applications have been proposed by different scholars. Because of the innovation and technology disruption of financial services by nonfinancial enterprises, with the help of fintech, customers can participate in a variety of mobile environment services—e.g., online payment, funds transfer, loan application,

purchase of insurance policies, management of organizational assets and management, stock investment, mobile payment, P2P lending, crowdfunding, and cryptocurrency [2].

The present study, which draws from the literature, supports Leong's [1] definition of fintech as a multidisciplinary subject that combines innovation management, technology management, and finance.

Digital transformation involves organizations or countries adopting new digital technologies, including disruptive innovations like social media, mobile, big data, cloud computing, IoT, AI, fintech, blockchain, virtual reality, and augmented reality, to enhance their performance significantly. This process requires a well-defined strategy, an appropriate organizational structure, digital capabilities, a supportive culture, and effective governance [3]. The widespread use of digital technologies in banking operations and services, incorporating various digital tools such as computers, computer networks, digital communication, the internet, and information and communication technologies with suitable software, results in heightened speed, security, and efficiency, offering numerous benefits to both banks and their clients; additionally, there is a proposed method for effectively transitioning a traditional financial data bank into a digital counterpart using a typical commercial/retail bank as an example [4]. Analyzing the digitization of banks requires taking a view that is centered on changing the way that this industry currently manages, including new worldviews that have arisen in the digital era in addition to management practices, technologies, processes, and tools [5].

An analysis of several applications, such as payments, advisory services, financing, and compliance, provides even more context for the fintech scene. As in previous studies on digitalization in banking operations, the research addresses the growing interest in digital transformation within the banking sector and emphasizes the need for a well-defined strategy, organizational structure, and digital capabilities to enhance performance [3–5].

In Romania, bank digitalization increased significantly in recent years, and according to FinnoScore, Romania has a bank that sits among the top three worldwide in relation to digitalization [6]. We notice an increased interest in a small part of the population in also using fintech accounts; according to a study, 6 out of 10 Romanians are using fintech applications [7]. On the other hand, cryptocurrencies' use is still low in Romania in comparison with developed countries. In this study, we aim to discover the factors that influence people in Romania to use financial products through online channels, determine whether there is any appetite for fintech accounts or even cryptocurrencies, and uncover how the replacement of physical cash with account money and online fast payments is perceived. With this aim in mind, we conducted quantitative research using an online questionnaire that was distributed to a variety of individuals with the aim of reaching a minimum number of 100 responses.

This number was calculated to obtain a 10% error margin and a 95% confidence interval for the sample size given the target population of Romanians—20–79 years old, residing in both urban and rural areas. The minimum target size for the sample is 97, assuming the presence of these restrictions.

The questionnaire consists of 34 questions, out of which 6 are related to demographic characteristics, such as gender, age, level of education, living location, income, and current professional status. The others relate to the participants' relationships with banking, type of services used, how they are used, preference of physical or online banking contact, intention to use other banking products in the future, trust in the banking system or other financial entities that are nonbanks (fintech companies), and preference for account payment rather than physical cash.

Our research objective is to analyze the factors that influence the decision to use or not use fintech accounts in Romania, and what products are of interest currently to the population who have completed tertiary education in the context of already using online/internet banking services and other banking products. Understanding these factors will help financial institutions invest in the development of their products' base and target certain types of customers. Additionally, we present certain factors that determine the

current use of online banking products and what the current perception of cash–noncash utilization is in the economy.

The sample comprises individuals that come mainly from urban areas and are graduates of tertiary education, as we wanted to gather feedback from more educated customers with regard to online financial services. Among people with tertiary education, fintech companies and online access to financial services are still not fully utilized; therefore, we believe that their answers are most relevant for our study in comparison with those of other demographics.

Placing Romania in the context of the European Union in relation to digital skills, we can see that the population of the country has the lowest degree of digital skills in comparison with the other countries included (Figure 1).

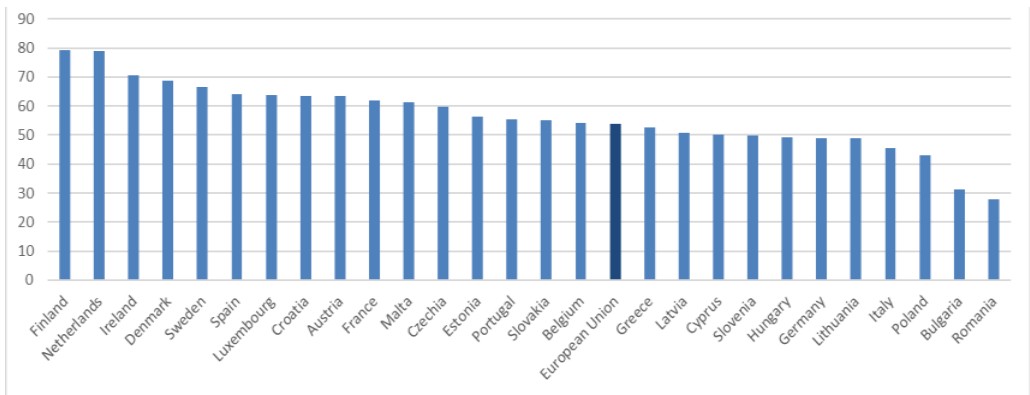

**Figure 1.** DESI indicator 2022 of a minimum of basic skills attainment among all individuals (aged 16–74). Source: the Digital Economy and Society Index (DESI) (authors' representation) [8].

We analyzed the DESI indicator 2022 [8]—basic digital skills attainment as a minimum—for individuals with 'basic' or 'above basic' digital skills in each of the following five dimensions: information, data literacy, communication and collaboration, problem solving, digital content creation, and safety. We assume an electronic banking user needs to have at least basic digital skills according to the DESI definition.

The specific research question we address in this paper is the following: What are the determinant factors behind the use of fintech services?

By analyzing the results of our distributed questionnaire, we notice that people in general are using online contact with financial entities. Account opening is preferred if online opening is possible. However, the second option for choosing their bank is the position of the bank in the top 10 of the banking system and only the third one is related to interest paid. Moreover, elderly people still prefer physical relationships with banks, pay with cash for various services (hairdresser, housecleaning, massage, childcare, restaurants), do not care if banks can provide virtual cards and would not be interested in financial services provided by fintech companies.

Furthermore, we use logistic regression to achieve insight into the key determinants and factors influencing the following: the adoption of financial products through online channels in Romania; the potential interest in fintech accounts and cryptocurrencies; perceptions of the replacement of physical cash with account money and online fast payments, particularly among the urban population with tertiary education.

The main contributions we bring to the literature in the field are related to understanding Romanian consumer behavior, digitalization trends in Romanian banking, perceptions of cashless transactions, and educated customer insights.

The paper "fintech Adoption Factors: A Study on an Educated Romanian Population" is structured as follows: Introduction—sets forth the background for the investigation, its goals, and its significance; Literature Review—provides a thorough analysis of the body of knowledge regarding the adoption of fintech and associated aspects; Materials and Methods—explains the research approach, including the procedures used for data

collection and analysis; Results—presents this study's conclusions based on the data gathered; Discussion and Limitations and Future Work—interprets and discusses the results in light of the research objectives and the pertinent literature, and identifies any study limitations and suggests potential directions for future research; Conclusions—summarizes the major findings and their implications.

## 2. Theoretical Background

The concepts of trust and distrust in the context of financial institutions create a critical duality in the landscape of internet banking [9]. Trust is in direct opposition to the fears associated with mistrust, embodying potential concerns [10]. Trust refers to the confidence in the system generated by these organizations. The adoption of online banking is the result of a complex decision-making process that is influenced by several variables, including the desire for change, level of trust, educational background, and others [11]. According to Chou and Chou [12], online banking has five main purposes: keeping track of account balances, viewing transaction history, paying bills, moving money between accounts, and accessing credit card advances or ordering checks.

Technical factors that influence user perception and adoption behavior include internet speed and the security of the online banking infrastructure [12]. According to technology acceptance theory, factors such as enjoyment and flow are crucial in illuminating a given person's technological experiences. An analysis of factors influencing e-commerce adoption shows that external pressures, perceived benefits, and readiness are the most important factors [13]. Interestingly, despite significant investment in new information technologies, the United States has often ignored the fundamental elements of technology adoption, such as planned change, internalization, trust, and the dynamics of adoption. Interestingly, innovators, who typically exhibit characteristics such as high wealth, education, active social participation, and leadership, are often the first to adopt new services or goods, such as internet banking [14]. The perception of potential benefits, which significantly influences the acceptance of service, is an important consideration in individual decision making about technology adoption [15]. Aside from personal preferences, factors such as consumer demand, market competition, technology accessibility, and overall market forces influence the adoption of online banking services among banks [16].

Montazemi and Qahri-Saremi [17] used a grounded theory approach for their literature review and the two-stage random effects MASEM procedure for their research; they identified the factors that influence the preadoption and postadoption of online banking. A study on the adoption of internet banking in China also found that security was the most important determinant of user adoption, with perceived ease of use and privacy policies also having a major influence [18]. Another study, conducted between 2003 and 2006 in the United States among a panel of commercial banks, examined the factors that led banks to use transactional websites (the precursor to online banking) for their customers. This study showed that, while the characteristics of individual banks (such as reputation and public trust) play a role, the banking sector is the area that the bank should primarily focus on, and competition is a crucial factor to consider [19].

Numerous advantages that internet banking offers over traditional banking have been identified through extensive research across the field. The most prominent of these is the extreme convenience of online banking, which provides users with constant access to their accounts and allows for transactions to be made from anywhere [20]. The trend towards internet banking, which not only reduces the need for physical bank branches but also benefits both financial institutions and customers, has been actively promoted by banks in the United States [21]. Online banking is attractive because it allows for 24-hour payment processing from the comfort of one's own home, eliminating the need to visit a physical branch [22].

However, some people are reluctant to use internet banking because they do not trust it. A market study conducted by Salus and Weeks shows that the lack of human interaction can be perceived as a problem; in its absence, customers are lacking the trust

they experience when visiting a bank branch, as this trust is absent in interactions within online banking applications [23].

In addition, research that considers cultural factors, such as the framework presented by Hofstede [24], has shown the importance of collectivism and the insignificance of other factors in predicting behavioral intentions and online banking usage patterns [25]. The effectiveness and acceptability of retail online banking has been studied, and criteria such as accuracy, security, network speed, ease of use and convenience have been shown to be important factors in perceived effectiveness [26].

Comparative studies comparing the adoption of digital banking in the United States and other growing economies, such as Russia and China, show that the US banking system has a clear lead in this area [27]. Economic value, ease of use, social influence, company reputation, features, and rewards all have significant impacts on customers' desire to use e-banking services exclusively; such findings have been reported by studies with generations Y and Z in Indonesia [28].

Zagalaz Jiménez and Aguiar Dáz [29] examine the relationship between parameters such as income and employment status and the use of internet banking in Spain. They find interesting correlations with these variables. Similarly, Szopiski [30] uses econometric studies to identify critical factors that influence the use of internet banking, such as access to the internet, trust in banks, and the availability of new financial products.

Ramayah [31] used discriminant analysis to examine survey data from Malaysian bank customers to find out which characteristics are most likely to influence the acceptance of e-banking. Vinayek and Jindal [32] use discriminant analysis to identify the factors that influence Indian customers' preferences for internet banking services. Lawson and Todd [33] analyze data from New Zealand using exploratory factor analysis to identify the stimuli that influence preferences for particular payment systems and to assess the influence of demographic and socioeconomic factors.

One study examines whether the adoption of digital payment methods, such as mobile payments, increases risks of financial vulnerability. It examines this relationship beyond the US, exploring the willingness among populations to use social media companies for money transfers and to share account information with third-party financial services, based on data from a Norwegian adult population sample (n = 2202) [34]. In contrast to findings from research in the US, mobile payment users were found to be less financially vulnerable than nonusers, with women being more willing to use digital payment technologies than men. Financial vulnerability was more pronounced among younger generations and those with less financial education but was not directly related to mobile payments or other digital payment methods.

Another study used planned behavior to examine how new technologies are adopted. A total of 118 employees participated in this study over a five-month period to determine how and the extent to which age influences the adoption of new technologies [35].

Defining the fintech framework, including the most appropriate business models for each country, proves to be the central dilemma of most studies in this area and is a mandatory step in creating regulations that can keep up with technological advances. Cybersecurity and information infrastructure must be constantly improved to support the large amount of data, some of which is personal in nature, used in fintech businesses. Interesting technologies such as robot advisors, big data, and AI can be supported by developments in the internet and smartphones [36].

According to a study conducted in Malaysia, published in an article by Rahim et al. [37], fintech contributes to financial inclusion by providing unbanked and underbanked consumers—especially low-income households and minority groups—with access to affordable and convenient financing to improve their economic opportunities. Fintech services reduce costs, improve the quality of financial services, increase employment rates, reduce poverty through lower transaction costs; this facilitates everyday personal and professional life to thrive and provides financial access through microfinance and crowdfunding.

Digital literacy and consumer skills can also be improved through technology in financial services. Fintech services reduce energy consumption (e.g., fuel) and increase environmental protection (e.g., carbon emissions). Although the concept is sound, these benefits may not, in reality, be realized, as the adoption of fintech services is low. Fintech is associated with cyber-related risks, which can be broadly categorized into loss of privacy, compromised data security, increasing financial losses due to fraud and scam, unclear legal status, lack of regulation, and the risk that fintech providers lack operational efficiency. Driven by financial technology (the origin of the term "fintech"), such as blockchain, big data, machine learning, and artificial intelligence (AI), fintech has enabled consumers to conduct financial transactions without the physical presence of people, money, or infrastructure. Fintech and its associated technologies provide digital solutions for affected individuals, businesses, and governments. At the height of the COVID-19 pandemic, governments' measures to contain physical movement and promote safe physical contact led to a massive adoption of fintech.

According to an Ernst and Young study [38]—"Global FinTech Adoption Index 2019"—the proportion of users utilizing fintech services, expressed as a percentage of the digitally active population for 27 selected countries, was 64% in 2019. It is worth noting that this figure is 87% for China, 71% for the UK, 64% for Switzerland, and 46% for the US. As shown by the results of the survey conducted by Capgemini and Efma [39], covering six regions of the world, millennials (people born between 1980 and 1995—as an age category, these people are presently 27–42 years old) are more likely than other generations to use the services offered by fintech. In our study, the 28–43 age category received the most responses, and we also find that this age category has the greatest interest in using smartwatches. In our study, the high-income categories also have the greatest interest in adopting fintech services.

For example, it can be seen that, in Western Europe, the adoption of fintech among Millennials is 65.6%, while in the other age groups it is 53.2%. In Central Europe, the difference between the acceptance of fintech among members of the millennial generation and the other generations was slightly lower at 9.5% (the corresponding figures were 72.6% and 63.1%, respectively). According to the results of the Ernst and Young report [40], the fintech adoption index for the millennial generation is 48% worldwide and 59% in the US. In Poland, this ratio was around 75% for the millennial generation in 2019 [41].

Das and Das [42] pointed out that 66.6% and 62.3% of people in the 18–28 and 29–39 age groups were regular users of fintech services, respectively, compared to only 26.9% in the over-50 age category. Similarly, Li et al. [43] found that younger consumers are more likely to use mobile payments than older consumers—the predicted probability of an adult aged 20 using mobile payments is almost 10 times greater than of an adult aged 75.

Gender is another demographic determinant that influences the adoption of new technologies. Research shows that men are more likely to adopt a new technology than women [42–45]. An interesting study on age was also conducted by Carlin et al. [46]. The authors investigated gender differences in technology adoption by analyzing a broader context, i.e., they examined the answer to the following question: how does better access to financial information through new technologies change consumer credit use and influence financial fitness? The results of the research show that men tend to be more inclined to adopt new technologies and access information to a greater extent than women; however, the economic impact of access is greater for women—each additional login to a smartphone application reduced bank fees by SEK 238.1 (USD 1.98) for women and by SEK 195.2 (USD 1.63) for men.

The research conducted by Rogers [47] in the context of technology adoption shows that early adopters may adopt a given innovation based on their higher level of education. This relationship was also analyzed by Szopiński [30], who proved that the level of education of respondents positively influences the use of online banking. Similarly, only a few of the studies conducted prove the influence of income level on the acceptance of technological innovations. The study conducted by Flavián et al. [44] on the influence of

income on the use of online banking shows that this factor has a significant influence on the acceptance of online banking services (a person with an annual income of more than EUR 36,000 was more likely to make transactions over the internet than someone with an income between EUR 24,000 and EUR 36,000 per year).

The likelihood of using innovative services offered by fintech decreases with age. Men are more likely to use innovative fintech services than women. The likelihood of using innovative fintech services increases with the level of education. The likelihood of using innovative level services increases with monthly net income.

Ryu [2] identified convenience as a factor related to portability and instant accessibility, offering consumers flexibility in terms of time and location. The author demonstrated that convenience has a positive effect on fintech adoption. The results of our research are consistent with this study and show that the more important factor is a person's ability to use a smartwatch; when choosing a financial institution, the more likely a millennial is to use a smartwatch, the more likely they were to use innovative fintech services. Demographic variables that have a statistically significant impact on millennials' use of innovative fintech services include age and gender. The results are consistent with those of other authors, such as Carlin et al. [46], Das and Das [42], Li et al. [43], and Liébana-Cabanillas et al. [48]. These two hypotheses have been clearly confirmed. The first is "the likelihood of using innovative services offered by fintech decreases with age". In their case, the probability of using innovative services offered by fintech decreases by 5.7% when the age of a respondent increases by one year. While the probability of a 25-year-old person using fintech is 64%, it is only around 53% for a 15-year-old person. Younger people are therefore more open to modern technologies. As Hanna and Kim [44] argue, it is possible that older generations are more anxious when trying to learn how to use mobile payments, which is associated with a lower perceived ease of use. In terms of gender, our research proves that men are more likely to use innovative services offered by fintech than women.

Accounting for age, gender, income, and education level, this study expands our knowledge of fintech adoption. This initiative builds on the findings of previous research that younger, better educated, and wealthier people are more likely to adopt innovative fintech services. The research findings have implications for financial institutions looking to tailor their offering to specific customer segments in the dynamic market for digital financial services [3,42,47,48].

## 3. Methodology

### 3.1. The Questionnaire

We used a questionnaire which can be found the end of this article, in Appendix A, to obtain data for our analysis. The questionnaire was distributed between August and December 2022, mainly online—on a LinkedIn profile and through email. We decided from the outset to target the questionnaire to educated people living in urban areas, as we assumed that they would have more information about online banking practices and tools, including potential fintech tools, and be more willing to use them. Therefore, using LinkedIn as the platform for collecting responses seemed the most appropriate, as this platform is mainly used by professionals from different fields; thus, the respondents were more likely to be within our required demographic. We received 118 valid responses. The response rate within the population that had access to the specific questionnaire was around 9%.

For a small proportion of respondents (those not using a computer), we used paper forms which were distributed manually. These were 10 respondents, representing 9% of the total number of responses returned. All responses are anonymous. Before the survey was distributed, it was sent to six economists who are familiar with the above questions; we improved the questionnaire through three iterations based on their feedback.

The survey is structured as follows: (i) demographic data; (ii) general use of internet-based applications in daily life: (iii) banking products; (iv) trust and use of online financial services; use and potential interest of fintech services and cryptocurrencies. Appendix B con-

tains the questions and brief descriptive statistics for each of them: number of valid answers, mean, median, mode, frequency of mode, standard deviation, skewness, and kurtosis.

Our sample group consists of 53% men and 47% women. Their age distribution is compared in Table 1 with the distribution of the Romanian population aged between 20 and 79 years old.

**Table 1.** Romanian population age groups—percentage of total.

| Age Group Romanian Population | 20–29 | 30–44 | 45–59 | >60 |
|---|---|---|---|---|
| % of total Romanian population | 14% | 28% | 29% | 29% |
| Age group sample | 18–28 | 29–43 | 44–58 | >59 |
| % of total sample | 9% | 53% | 33% | 5% |

Source: National Statistics Institute data, 2021. [49].

According to official figures, total population of Romania in 2021 was 19,053,815 people [49]. Of these, 14,083,368 were between 20 and 79 years old, while there are 3,056,898 people with a tertiary education. Of the tertiary-educated population, 45% are women and 55% are men.

We used a *t*-test to calculate with 95% confidence whether there are differences between the structure of the Romanian population and our sample. The 2-sided statistical value of the *t*-test with 3 degrees of freedom is 3.182, while the value calculated by our *t*-test is 1. We accept the null hypothesis that the above distributions (the Romanian population and the sample) are not different.

*3.2. Logistic Regression*

A logistic regression model (LR) was used to analyze the collected responses. These models are frequently used in various scientific fields, especially in epidemiology, to study the effects of risk factors on disease development [50]; however, they are also often chosen as a method for economic modeling. For example, LR has been used to determine the variables that control the customer's binary decision to use or not to use electronic banking services [51] and mobile banking services [52]. In addition, logit regression can also be used to determine the factors that influence a bank's decision to adopt online banking [53]. Logistic regression is useful to determine the probability of the presence or absence of a particular characteristic, or whether a person belongs to a category; this is similar to discriminant analyses. For example, Tesfom and Birch [54] applied logistic regression to determine the barriers that prevent young and old bank retail customers from switching to a new institution.

Logistic regressions can be used in qualitative choice models to determine the probability that an individual will make a particular choice or exhibit a characteristic based on a vector of explanatory factors. A qualitative choice model requires a finite set of alternatives from which a mutually exclusive choice must be made. If there are only two alternatives to choose from—typically a positive choice and a negative choice—and the choice follows a logistic distribution [51], then we the LR used to estimate the probability of the outcomes a would be referred to as a binary logit model. The cumulative distribution function of a logistic distribution, the logistic function, has the form of the well-known S-shaped sigmoid curve. Equation (1) shows the regression formula, where y is the dependent categorical variable, $\alpha$ is the function constant, e is the error term, X is the vector with the independent variables from $x_1$ to $x_n$, and b is the vector containing the slope of the parameters from $\beta_1$ to $\beta_n$.

$$y = \alpha + \beta x + \varepsilon y = \alpha + \beta x + \varepsilon \tag{1}$$

In comparison to ordinary least square regression (OLS), LR models show a nonlinear relationship when the criterion values are probabilities or odds and a linear relationship if the criterion values are log–odds, also known as logits. Logits can range from negative to positive infinity and correspond to the coefficients estimated through OLS [55]. If

the predicted variable of the regression is dichotomous, this variable corresponds to the predicted probability that the variable has a value of one. Applying OLS to a categorical and binary response variable means that the curvilinear relationship between each predictor and the output variable is not reflected and that the constant incremental effect that a change in a predictor has on the outcome is not preserved. For this reason, OLS provides the best-fitted results when the variable being regressed on is continuous [55].

Referring to our research question, related to the use of fintech services—a notion represented by a categorical response variable—LR is better suited to test our hypothesis than either OLS or discriminant analysis because we use categorical variables that do not follow a normal distribution [55]. As mentioned before, logistic regression is adequate when attempting to study the effect of a set of predictor variables on a categorical qualitative outcome [50]. One interpretation of LR would be that, while the linear OLS regression coefficient represents the constant rate change across all the values of the predictor, in probabilistic LR, the rate of change will vary based on the value of the predictor; here, there are smaller values at the end of the function tails than near the middle, leading to a weaker relationship at the extremes [55]. To address this issue, a constant representation of change can be modeled by using odds ratios and log odds ratios. Odds are equivalent to the ratio between something occurring and something not occurring [55,56]. It is important to note that odds and probabilities follow the following relationship:

$$Probability = \frac{Odds}{1 + Odds} \tag{2}$$

Logit regressions quantify the relationship between the predictor and response variable through odds ratios; this is the ratio between the odds of one option being selected over its alternative or the odds of a trait being present over the odds of it not being present [50].

To demonstrate the statistical significance of LR results with a large sample, we can calculate the confidence interval for the natural logarithm of the odds ratio using a chi-square test, as the shape of the probability density function of a logistic distribution approaches that of a normal distribution. The log–odds of LR, also referred to as the "logit", is a close equivalent of the beta coefficient estimated by an OLS, representing a linear relationship between the explanatory and response variables. The predicted and observed outputs are compared by maximizing the log–likelihood function to reach the best fit [53]. Therefore, the null hypothesis in logistic regression is that the slope of y over x is equal to zero and that log of the odds of x being present is not linearly related to y. The log of the odds ratio is then treated as the coefficient of a linear relationship, indicating the strength of the explanatory variables' effect on the response variable. Because y is a discrete variable that can only take values of 0 or 1 (corresponding to S occurring or not), the coefficient of x controls the probability of S occurring [50].

### 3.3. Model Preparation and Assumptions

Creating a logistic model where the predictor variables are discrete and binary, nominal, or ordinal is simple and requires little setting up. For binary variables, we can simply set up dummy variables with values of 0 for the absence of a trait or negative response and 1 for the presence of a trait or positive response [50]. In the case of a nominal predictor that has more than two states, we would need to construct several dummy variables to represent this predictor equal to the number of categories subtracted by one. For ordinal predictors, this method of nominal coding can also be used when there are up to 5 states or ordinal coding can be used, in which case we construct a variable whose value is incremented to represent successive intervals including the increasing values of the predictor [50].

We have decided to introduce the variables presented in Table 2 on the basis of the survey questions. Our response variable is based on Question 30 and shows an individual's choice to use fintech services. Nominal variables are coded as 1 for "no" and 2 for "yes", apart from gender (Gen), where 1 indicates a man and 2 indicates a woman. Age and income (Inc) are coded using a scale from 1 to 4 (from younger to older ages), and 1–5 (from

low to high). Tec5G, e-wallet, and Vcard are scaled from 1 to 5 (from irrelevant to highly relevant). We have chosen the variables presented in Table 2 considering the following considerations: demographics (gender, age, annual income), experience or familiarity, trust in other internet-based applications (mobility applications, smart watch, trust in 5G technology), previous financial investments, and modern card features (electronic wallet and virtual cards).

**Table 2.** LR model preparation.

| Dependent Variable | | |
|---|---|---|
| *Question* | *Category* | *Type* |
| Do you use financial services by fintech companies? (Q30) | fintech | Nominal |
| *Independent variables* | | |
| *Question* | *Category* | *Type* |
| What is your gender? (Q1) | Gen | Nominal |
| What is your age? (Q2) | Age | Ordinal |
| What is your annual income? (Q6) | Inc | Ordinal |
| Do you or your family use mobility applications? (Q7.4) | MobApp | Nominal |
| Do you or your family use Smartwatch? (Q7.6) | SmartW | Nominal |
| Do you use other financial services (Q8.6) | Inv | Nominal |
| Do you trust 5G technology to help the development of financial services? (Q11) | Tec5G | Ordinal |
| With regards to your card, how important is it for you to use it in the electronic wallet? (Q26) | E-wallet | Ordinal |
| With regards to your card, how important is it for you to issue a virtual card in your online/mobile banking application? (Q28) | VCard | Ordinal |

Source: Authors' work.

The way we set up our data fulfills the first two assumptions of LR: the dependent variable is dichotomous; there are one or more independent variables. Because each observation represents a different respondent, the condition of observation independence is also met. We used SPSS software V28 to estimate the parameters of LR.

One assumption that needs to be fulfilled is the lack of influential outliers in the dataset. We computed Cook's [56] distance and estimated the absolute studentized residuals for each observation. We have decided to use Cook's original cutoff ceiling of 1 because the median of an F distribution with 9 (number of variables) and 109 (number of observations subtracted by the number of variables) is close to 1 [0.932]. For the studentized residuals, we applied a threshold of over absolute 2 [57]. Under those criteria, none of the parameters were excluded on the premise that they were influential outliers (the estimated values are found in Appendix C).

The advantage of logistic regression as a generalized linear model is that it bypasses most of the assumptions that constrain linear models. For example, LR does not require linearity between the dependent and independent variables and does not require the observations to be normally distributed [58]. For continuous data, checking for linearity using methods such as the Box–Tidwell [59] test is still useful. However, a binomially distributed error is one of the assumptions required to be true in LR [60]. Probabilities must also be a logistic function of the explanatory variable. Dependent variables must also not be linear combinations of each other [58], to avoid the multicollinearity problem.

It is commonplace to triage the variables that will be used to construct a representative model using a variable selection method. For example, in Laukkanen and Pasanen [52], a backward stepwise method was involved in the variable selection process, whereas Tesfom and Birch [54] opted to use forward stepwise selection. We use forward stepwise variable selection to select the optimal set of predictors by progressively introducing additional independent variables into the model until the best fit is reached. The procedure starts with a base intercept; this is the only model that disregards any predictors. At each step,

we conduct a likelihood–ratio test between the simpler model and a model containing one more variable from the set of unused variables. We then set the base model and calculate the log–likelihood of the LR. Based on the results of the LR test, we adopt the model that exhibits the lowest *p*-value at each step. This process helps us to introduce as many statistically important variables as possible without diminishing the R2 value of the final model [58]. The threshold *p*-value for variable selection is 0.05.

## 4. Results

First, we analyze the sample group starting from the demographic characteristics: age, annual net income (result is presented in EUR at the calculated EURRON rate of 4.92), gender, level of education completed, professional status, and living location. Second, we present the results of the LR model.

### 4.1. Brief Analysis of Responses

Most respondents to our survey are male (53%), all of whom have tertiary education and 68% of whom have postgraduate studies. A total of 53% of the respondents fall in the age category of 29–43 years; 97% of the respondents live in urban areas, 82% of the respondents are employees, 7% are entrepreneurs, and 6% are freelancers. Most of the respondents (53%) have an annual net income above EUR 24,186.

The age category 18–28 in our questionnaire is large enough to cover young people with tertiary studies as well. We had only three answers in this range, and all of them have post-university studies, which is possible in Romania; bachelor studies take 3 years, master's studies take 2 years, and university usually starts at around 18 years of age.

Analyzing the internet utilization for various applications among the sample we found that people below 58 years old are using either TV applications (see Figure 2), mobility applications (see Figure 3), cryptocurrencies (see Figure 4), or smartwatches (see Figure 5), or in various proportions; meanwhile, the category older than 59 years has the lowest interest in using such products. Figure 1 shows that 100% of participants in the 18–28 years age category used TV applications and mobility applications; in contrast, only 33% of participants in the age category >59 years old use such applications. We also see a reduction in use of these applications together with age increase, as 90% of the age category uses TV applications, which drops to 87% among the 44–58 age category, who show an 85% rate for mobility applications. Cryptocurrencies are not used at all by the 18–28 years and >59 years age categories, probably because young people do not have the available funds to invest in such applications and older people do not trust them. Only 19% of the respondents aged between 29 and 43 and 13% aged between 44 and 58 were found to be using cryptocurrencies. Being more accessible, smartwatches have a higher rate of use in the sample population. However, the same trends were seen with smartwatches as were seen with the use of cryptocurrencies: there was no interest in them among the older generation (>59 years age), and the highest interest in them was found among the 29–43 years age category—68%.

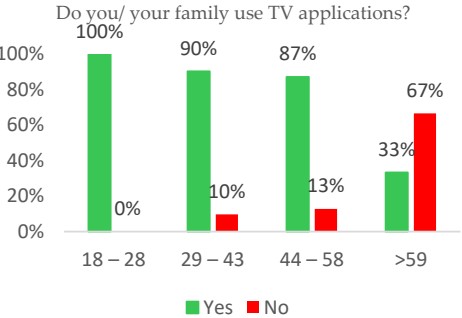

**Figure 2.** The use of internet-based TV applications. Source: authors' representation.

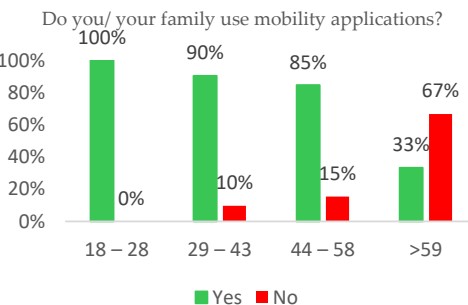

**Figure 3.** The use of internet-based mobility applications. Source: authors' representation.

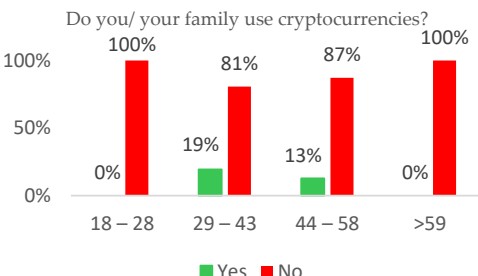

**Figure 4.** The use of cryptocurrencies. Source: authors' representation.

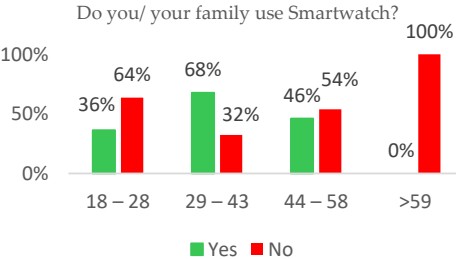

**Figure 5.** The use of smartwatches. Source: authors' representation.

When referring to the types of banking products our sample respondents use, 97% have current accounts, 94% use payment by card at merchants, 65% have credit cards, 91% are using payments through online/mobile banking applications, 58% have outstanding personal needs/car/leasing loans, and 57% have outstanding investments in state bonds and mutual funds. If respondents had savings above EUR 10,000, then they would choose the following utilization of the funds: 48% would make financial investments, 26% would make bank deposits, 14% would spend the money in personal development, 4% would spend it in house renovation, 3% would buy a car, and 3% would keep the money at home (see Figure 6). Split by the value of annual income, we notice that the higher the income, the higher the percentage of participants would choose to place it in financial investments. A total of 60% of the people with an annual income above EUR 24,186 would place their savings in financial investments, while only 27% of people with the lowest income would place their savings in such investments. People with low income (47% out of participants with annual income below EUR 6098) chose the safest products, such as bank deposits; the higher the income, the less participants chose bank deposits (only 22% out of the sample with income above EUR 24,186 would choose bank deposits). We also observe that people with an average income (between EUR 6098 and EUR 24,186 annual income) would spend their savings on personal development, in a proportion that ranges between 50% and 29%. The choice of personal development is positive for society and progress. Choices below 5% of the sample size are not relevant, although they are more targeted toward spending alternatives.

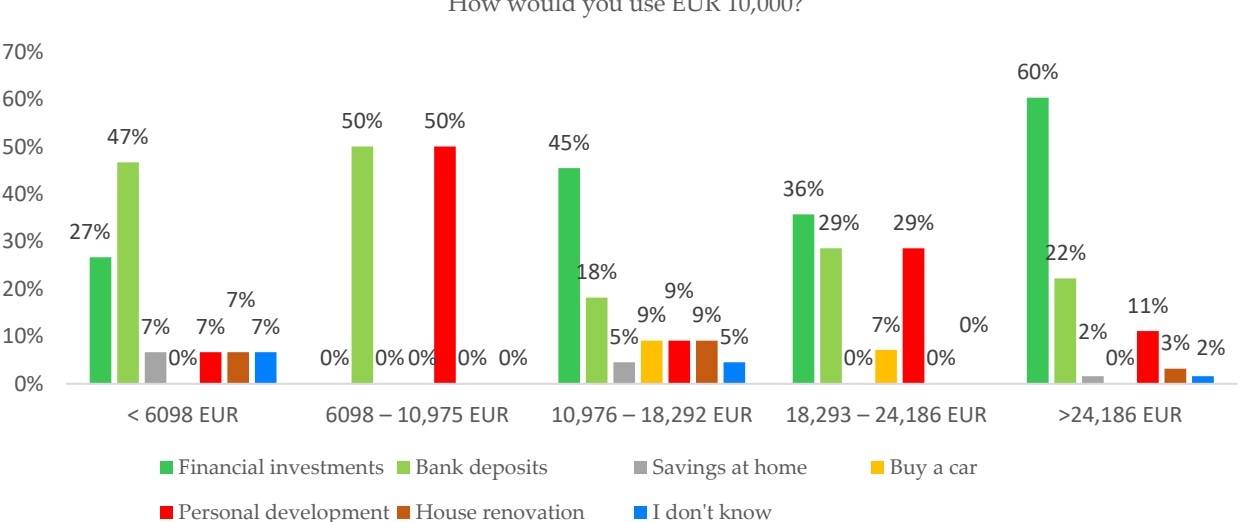

**Figure 6.** Investing/spending money. Source: authors' representation.

Out of the sample group, 16% are working with one bank only, 36% with two banks, 31% with three banks, and 17% with four or more banks. With respect to the first criteria of choosing their bank, 55% of the respondents preferred banks that offered the option of opening online accounts, 25% selected the bank based on physical distance, 10% selected based on image (top 10 by assets), and 9% selected in accordance with recommendations from friends—see Figure 7. Analyzing the choice for account opening by income group, we see that the highest percentage of online preference comes from the highest annual income—66% out of the sample category would choose the online option. We also notice that the lower the income category, the higher the percentage of participants choosing a physical alternative (50% out of the respondents with income EUR 6098–10,975). We also see that friends' recommendations would play a good role in the case of the middle-income category.

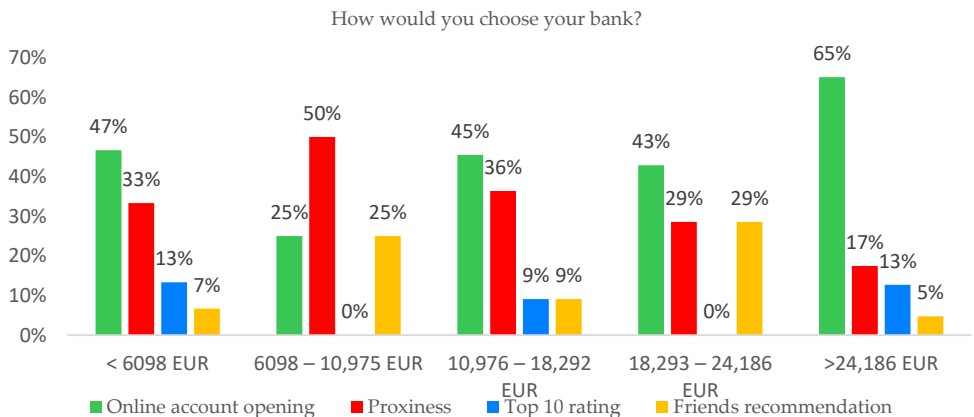

**Figure 7.** First option in account opening with banks depending on income. Source: authors' representation.

Most survey participants (79%) do not plan to use a new credit card in the following year, while only 9% plan to use one. In addition, 75% of the participants do not plan to take a new loan in the following year, while only 13% plan to take one. With respect to direct interactions with bank branch officers, 53% of the respondents stated that such interactions occur a maximum of once per year or not at all; 42% said that such interactions occur only a few times a year; only 5% state that such interactions occur once or several times per month. Regarding direct interaction with the banks' call centers, 49% of the respondents stated that

such interactions occur a maximum of once per year or not at all, 47% said that that such interactions occur only a few times a year, and only 3% stated that such interactions occur at least once per month.

Regarding the possibility of using existing cards in mobile wallets, 14% of respondents consider it to be unimportant or that they do not care, 71% consider that it is important, and 14% are indifferent. The cardless feature (the possibility to withdraw money from an ATM without a physical card) is important and very important for 52% of the respondents, while 33% of the respondents are indifferent to this possibility. In addition, 15% of the total number of respondents stated they did not need it. Moreover, when asked about the possibility of issuing a virtual card through an online application, 25% of respondents were indifferent, 11% were not interested in using it, and 64% consider it important or very important.

Concerning the modality of opening bank accounts, 15% prefer to undertake this task physically with direct interaction, 72% prefer online, and 12% are indifferent. With regard to the modality of making loans or deposits with banks, 23% prefer to undertake this task physically with direct interaction, 65% prefer online, and 10% are indifferent. Finally, the ESG (Environmental, Social, and Governance) actions of their bank were important or very important for 65% of respondents, while 35% were not interested or indifferent.

Online or mobile banking was used by 96% of the respondents, and 80% of them discovered new features/products/services when using the online application. In addition, 96% of internet banking users connect to the online banking application using smartphones, while 49% also use laptops, and 14% use desktop computers. Moreover, when asked about the frequency of use of mobile banking (accessed through smartphone or tablet), 10% of the respondents stated that they do so a maximum of once per year or not at all, 6% prefer to use it only a few times a year, and most—84%—said that they use it at least once per month or several times per month. When asked about accessing online banking through a desk PC or laptop, 26% of the respondents stated that they access it this way a maximum of once per year or not at all, 25% reported this mode of access to occur a few times a year, and 48% stated that this kind of access occurs at least once per month or several times per month.

Overall, 85% of survey respondents report trusting banks in Romania that are under National Bank of Romania (NBR) surveillance, whereas only 14% do not trust these banks. A total of 85% of the respondents would also trust the 5G capabilities to help develop financial products.

The last part of our survey refers to fintech accounts; we enquired about use so far, potential interest for the future, and interest in cryptocurrencies. At the entire sample level, 72% of the participants declared that they were already using a fintech account. However, younger participants are more intensive users of fintech products than older respondents. People with previous financial investments tend to use more fintech products, namely 78% in our sample —see Figure 8. In addition, most respondents (70%) use a card for payments abroad, followed by rapid payments to friends (22%). Very few use fintech products to exchange currency (7%) or receive income from abroad (2%)—see Figure 9.

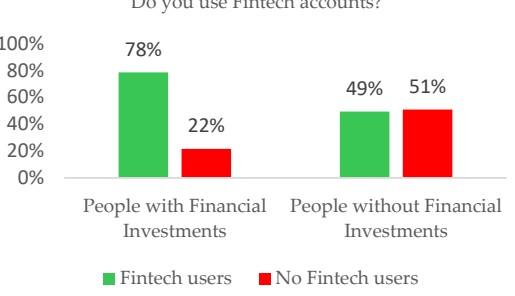

**Figure 8.** Fintech products use depending on having previous financial investments. Source: authors' representations.

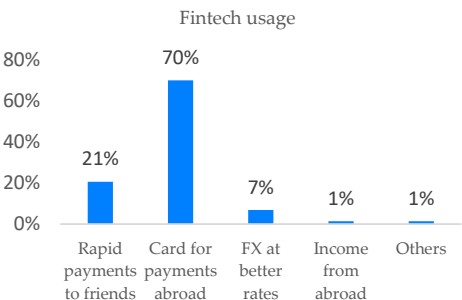

**Figure 9.** Fintech products use. Source: authors' representations.

A proportion of 70% of participants in this study reported that they would be open to having a fintech account, while 14% reported that they would not want such accounts. Interestingly, the highest proportion of respondents willing to use a fintech account comprised the second highest income category (29%); however, in most income categories, the percentage of respondents saying they would be interested in opening an account was around 25%—see Figure 10. With regard to the potential interest in using cryptocurrencies, 30% of respondents reported that they would not want to use them, and 46% reported that they would refrain from using them.

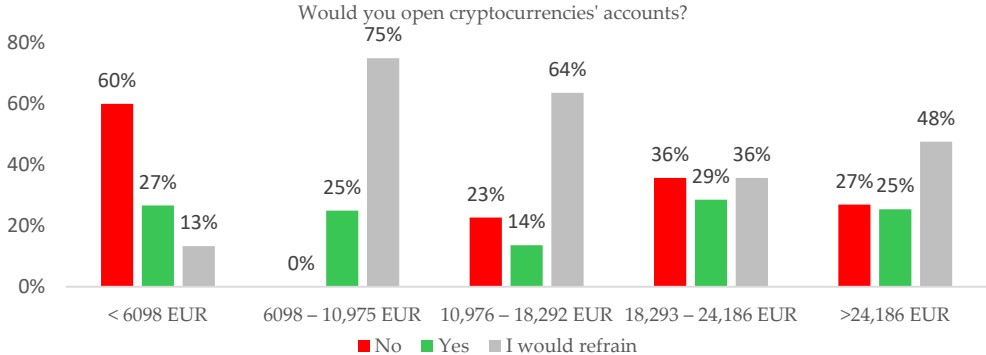

**Figure 10.** Interest in using cryptocurrencies. Source: authors' representation.

### 4.2. Logistic Regression Estimates

To test our hypothesis related to the use of fintech services, we first constructed a linear equivalent of our LR model to calculate the tolerance for the multicollinearity assessments using the variables in Table 2. We also computed the reciprocal of the tolerance which is the variance inflation factor (VIF). When square rooted, the VIF shows how much the standard error is inflated by multicollinearity. None of the estimated values shown in Table 3 would indicate levels of multicollinearity that could distort the estimation. As suggested by Hair Jr. et al. [58], we use a tolerance of 0.10, a floor for the cutoff, and a VIF of 10 as a ceiling for the cutoff. We further compared the variance proportions for the condition's indexes (relative eigenvalues) of the coefficient variance–decomposition matrix (see Appendix C) and failed to find any significant traces of multicollinearity among the independent variables [58].

After ensuring that the non-collinearity assumption is met, we fit both the null model and the full model using all variables in Table 2. We computed the likelihood–ratio test using the log–likelihoods of the two models with 9 degrees of freedom. The test statistic approaches a chi-square asymptotically, with degrees of freedom that are equal to the difference in the number of predictors between the complex and the simple models [61]. Based on the test results (Table 4), we reject the null hypothesis of the null model and the complex model explaining the response variable to the same degree [55]. Therefore, the complex model provides a significantly better estimation of the observed data.

**Table 3.** Multicollinearity diagnosis.

| Coefficient | Tolerance | VIF |
|:---:|:---:|:---:|
| Gen | 0.831 | 1.204 |
| Age | 0.861 | 1.162 |
| Inc | 0.709 | 1.411 |
| MobApp | 0.598 | 1.674 |
| SmartW | 0.815 | 1.227 |
| Inv | 0.712 | 1.404 |
| Tec5G | 0.944 | 1.060 |
| E-wallet | 0.569 | 1.756 |
| VCard | 0.597 | 1.674 |

Source: Authors' work.

**Table 4.** Likelihood–ratio test.

| $-2 \ln L_0$ | $-2 \ln L_1$ | $\lambda$ | df | $p$ |
|:---:|:---:|:---:|:---:|:---:|
| 156.875 | 131.544 | 25.331 | 9 | 0 |

Source: Authors' work.

We ran the logistic regression using a stepwise selection procedure that admits variables with $p$-values smaller than 5%. As shown in Table 5, the stepwise procedure went through two steps, henceforth referred to as S1 and S2, to consolidate the constant-only model (S0) and introducing the significant variables Inv and EWallet. The first variable to be rejected was age at a $p$-value of 0.086 (see Appendix C). The remaining variables improve the model's estimation even less and are therefore left out of the model.

**Table 5.** Model's fit (coefficients of determination).

| Step | $-2$ Log–Likelihood | Cox and Snell R Square | Nagelkerke R Square | McFadden R Square |
|:---:|:---:|:---:|:---:|:---:|
| 1 | 146.049 | 0.088 | 0.119 | 0.069 |
| 2 | 139.076 | 0.140 | 0.190 | 0.113 |

Source: Authors' work.

Table 5 shows the model coefficient of determination values in the form of $-2$ log–likelihood, also known as the model's deviance and the pseudo-R squared values. Being an unstandardized indicator, it is challenging to interpret the $-2$ log–likelihood. We rely instead on Cox's and Snell's [62] and Nagelkerke [63] coefficients of determination for measuring the model's ability to explain the variance of the response variable at each step. Nagelkerk $R^2$ has the benefit of being bounded between 0 and 1 [55]. McFadden's [64] coefficient is believed to be the best measurement of the model's accuracy because of its resemblance to the linear regression's $R^2$ [55,65]. The value of $-2$ LL dropped from S1 to S2, confirming that a more complex model has improved predictive power. Despite the pseudo-R squares also increasing from their S1 to S2, their values are quite low, implying that the model only explains a small fraction of fintech variability.

We complement the coefficients in Table 5 with an Omnibus coefficients test, which is a likelihood–ratio test that evaluates the overall fit of the nested model at each step by comparing the deviance of the observed values and the predicted probabilities. For this test, the null hypothesis is that the more complex model Si does not differ in predictive power from Si-1. Table 6 shows that each step increases the chi-square coefficient, progressively improving the fit of the model. The $p$-values are below the 5% confidence threshold, meaning that we can reject the null hypothesis.

**Table 6.** Omnibus tests of model coefficients.

|  |  | Chi-Square | df | Sig. |
|---|---|---|---|---|
| Step 1 | Step | 10.826 | 1 | 0.001 |
|  | Model | 10.826 | 1 | 0.001 |
| Step 2 | Step | 6.973 | 1 | 0.008 |
|  | Model | 17.799 | 2 | 0.000 |

Source: Authors' work.

A well-used goodness-of-fit test that assesses how close the model's predicted values are to the observed data is the Hosmer and Lemeshow [66] test. This method orders and divides observations in equal groups, and then checks for statistical differences between groups. However, the test has some serious drawbacks, mainly the requirement of a sample size of at least 400 observations, diminished fitting power for small samples, and oversensitivity to the number of selected groups. Because we used a small sample size with a reduced sample space, we cannot afford to use this method.

Allison [65] suggested running the Stukel [67] or other ungrouped goodness-of-fit tests in logistic modeling instead of the Hosmer–Lemeshow approach. For test samples comprising 100–500 observations, when modeling a linear function with two exogenous determinants and one interaction, Stukel outperforms a series of GoF tests, including the standardized Pearson test, the information matrix, and—to an extent—the Hosmer–Lemeshow test, in detecting interactions. The drawback is that the model performance visibly diminishes at lower sample sizes [66].

The Stukel [67] test involves comparing the baseline logistic model and a generalized version that includes two additional parameters that, depending on the fitted model's linear predictor being positive or negative, take values equal to either 0 or the squared linear predictor. The null hypothesis that the $z_a$'s and $z_b$'s coefficients are equal to zero is tested, where the values of the parameters are defined by Equation (3) and g is the linear combination of the model to be fitted.

$$z_a = f(g) = \begin{cases} g^2 & g \geq 0 \\ 0 & g < 0 \end{cases}$$
$$z_b = f(g) = \begin{cases} g^2 & g < 0 \\ 0 & g >= 0 \end{cases}$$
(3)

Table 7 shows the $\beta$ coefficient, Wald statistics and *p*-values for $z_a$ and $z_b$. Both probability values exceed the 0.05 threshold by a wide margin, indicating that we accept the null hypothesis that both $z_a$ and $z_b$ are nondistinct from zero. This indicates that our model adequately reflects the relationship between the covariates and the outcome variable.

**Table 7.** Results of the Stukel test.

| Coefficient | B | Std.E. | Wald | df | Sig. | Expβ |
|---|---|---|---|---|---|---|
| $z_a$ | 0.431 | 0.613 | 0.494 | 1 | 0.482 | 1.539 |
| $z_b$ | −1.023 | 1.579 | 0.420 | 1 | 0.517 | 0.359 |

Source: Authors' work.

Table 8 shows the contingency or classification table for observed y values and the values predicted by the model for S1 and S2. We can see that S2 marginally improves the model's ability to distinguish between fintech users and nonusers. It is interesting to note that, while the final model has an 80% accuracy in predicting fintech users based on Inv and eWallet, it has difficulties in distinguishing between nonusers, and its ability to do so has decreased from S1 to S2.

**Table 8.** Classification table.

| | | Predicted Fintech Values | | Percentage Correct |
|---|---|---|---|---|
| | | **1.00** | **2.00** | |
| Step 1 | Observed fintech values | 1.00 | 34 | 11 | 75.6 |
| | | 2.00 | 33 | 40 | 54.8 |
| | Overall Percentage | | | | 62.7 |
| Step 2 | Observed fintech values | 1.00 | 21 | 24 | 46.7 |
| | | 2.00 | 14 | 59 | 80.8 |
| | Overall Percentage | | | | 67.8 |

Source: Authors' work.

Table 9 contains the model-estimated coefficients and odds ratios for the final model (S2), which we will briefly discuss in this section. As mentioned before, the β coefficients or the logits are the natural logarithms of the odds ratios, which—in turn—are referred to as Expβ. While logits are the linear representations of the effect of the explanatory variable on y, Expβ represents the change in the odds of fintech; this has a value of 2 when we increase the independent variable by one unit (i.e., the odds of fintech = 2 when Inv = 2). A positive β and an Expβ higher than 1 would indicate a positive impact of the dependent variable on fintech. In contrast, a negative coefficient and a lower than 1 odds ratio would indicate otherwise. A Wald test statistic is used to check the significance of the coefficients. We can see that Inv has a robust positive effect on fintech at $p = 0.004$, whereas Ewallet has a moderately positive impact at $p = 0.01$. While Inv can only take values of 1 and 2, Ewallet has a range of values from 1 to 5; therefore, at its full extent, interest in the use of electronic wallets plays a more important role in the use of fintech than a previous tendency to invest in financial securities and equity. We can read the odds ratio of Inv as investors being 3.48 times as likely as noninvestors to use fintech services [66]; by applying Formula (1), we can estimate that Inv increases the probability of fintech = 2 by approximately 77%.

**Table 9.** LR model-estimated coefficients.

| Coefficient | β | Std.E. | Wald | df | Sig. | Expβ |
|---|---|---|---|---|---|---|
| Inv | 1.243 | 0.430 | 8.341 | 1 | 0.004 | 3.467 |
| Ewallet | 0.470 | 0.183 | 6.620 | 1 | 0.010 | 1.600 |
| Constant | −3.130 | 0.953 | 10.784 | 1 | 0.001 | 0.044 |

Source: Authors' work.

Because Ewallet is an ordinal variable, we estimated that a one-unit change in the variable's value will result in a 60% increase in the odds. Table 10 shows the probabilities associated with each of its value intervals [58].

**Table 10.** Probabilities of y = 2.

| Inv | | Ewallet | |
|---|---|---|---|
| **Values** | **Probability (%)** | **Values** | **Probability (%)** |
| | | 1–2 | 61.53 |
| | | 2–3 | 72.18 |
| 1–2 | 77.61 | 3–4 | 83.84 |
| | | 4–5 | 88.61 |

Source: Authors' work.

The rest of the variance is incorporated in the negative constant term, which indicates that, in the absence of previous interactions with investment and interest in an electronic wallet, an individual would default to not accessing fintech services. The variables that were not included in the model were not significant (the results are available from the authors).

## 5. Discussion

This section offers an interpretation of the findings in the previous section, as well as additional insight into the link between digitalization and financial products in a wider societal context for Romania. Thus, we discuss the relevance of gender, age, financial experience (such as investments), income, previous use of mobility applications, e-wallets, smart watches, virtual cards, and trust in mobile technologies.

First, we address our reasoning for introducing gender as a differentiator between users and nonusers of fintech in our LR model. Similar to Liao and Cheung [26], we believe that Romania has reached a development level where there would be no observable difference between genders in their perception of e-banking. This presumption does not necessarily hold true for all developed countries; for example, Jiménez and Díaz [29] found that men are more likely to use internet banking than females in higher education brackets, whereas the opposite has been shown in secondary studies. Our test results do not support gender as a factor that influences a person's decision to open a fintech account. In this sense, gender equality pervades the financial habits among our sample.

According to Anysiadou [68], demographic characteristics, such as age, combined with personalization and consumer risk perception play an important role in the adoption of virtual bank cards adoption. The youngest consumers are optimistic about their use, whereas the oldest consumers are negative about their use. The survey results show that most respondents below the age of 59 prefer opening bank accounts through online means (72% of the total sample) as well as accessing other instruments such as deposits and loans (66% of the total sample) digitally. At first sight, the survey data endorse the belief that the use of fintech services decreases with age, which persists in the literature. However, the results of the logistic regression in our model prompt us to reject age as a factor that influences decisions to open fintech accounts. The age variable meets the 10% significance threshold, but not the 5% one; therefore, it was excluded by the variable selection procedure. It is worth mentioning that, when Ewallet is removed as a parameter, age becomes significant, with a moderate negative coefficient. In light of the multicollinearity test results, we exclude the possibility of Ewallet being a substitute for age; in fact, we see it as a more accurate indicator of the propensity to use fintech services. It is surprising to see age evidenced as an irrelevant factor in our sample data. Our results seem to be at odds with other studies. It was noted that old individuals are deterred from using electronic banking because of their risk adversity and preference for conventional banking relations [51], even in the case of higher education graduates [29]. From research shared by Nawi et al. [69], we know that respondents' ages also have a significant negative effect on the tendency to resort to online purchases. This leads to a demographic disparity in the use of fintech services such as mobile payments [43] or peer-to-peer transactions [70], where younger users far outnumber older users. In some cases, due to their reluctance to use fintech services, specially tailored promotions have been proposed to attract older users [71]. It is also evident from the data collected that Romanian seniors are more reluctant to use smart TV applications, mobility applications, and smartwatches, all of which rely on digital technology. Taking these findings into account, we would have expected age to have a negative impact on fintech adoption. We consider the possibility that age plays a lesser role in the model due to the education level of respondents or due to the unbalanced distribution of the age groups of respondents in our sample. Given our results, it is uncertain whether age is a factor that should be considered when identifying fintech users in Romania.

The influence of income on the use of digital banking and financial services is a point of debate within the scientific community, but it is widely assumed that wealthier individuals are more inclined to use financial services. Higher-income bank customers have more disposable income and are more likely to use more of the financial features offered by banks to take advantage of the investment opportunities available to them. According to Lawson and Todd [33], married men over forty who are well educated and have a good income are more likely to use electronic payment methods because of their good financial status; meanwhile, those with modest incomes and moderate qualifications are more cash-oriented,

thrifty, and attracted to low-fee services. Jiménez and Díaz [29] assert that income has a positive influence on e-banking use; surprisingly, however, they also observed that, when high income and high education are both present, the level of engagement with e-banking decreases. This is possibly because those customers' financial needs include sophisticated financial assets that explain a closer, more involved, and direct relationship with the banks. In our case, we do not notice a tangible relationship between income and the desire to use e-banking services.

We know that a feature-rich and comprehensive banking application attracts new users. Windasari et al. [28] show that easy-to-read and easy-to-use e-banking is crucial in attracting and convincing young users to engage with digital banking. Rewards, such as cash tags or cashback, unique features, and positive word-of-mouth, are also important in delivering positive customer experiences and prolonging customer loyalty. Electronic wallets could be an entry point for consumers leading to future fintech services. An e-wallet should be the first feature introduced by a company looking to experiment with a fintech portfolio geared towards the younger generation. Older customers could be offered a different service package that excludes the digital infrastructure altogether. Successful examples are exhibited by banks in Poland [35] and New Zealand [33]; these have introduced an internet-free business model for older people, offering free deposits and cash transactions that can be processed by post—proving that this strategy is feasible.

The survey data, validated by the model, reveal that respondents who already have experience with investments are far more likely to also use a fintech service than those that do not hold such investments. Previous investing activities create a positive bias towards seeking fintech services. The attractiveness of a digital investment alternative is explained by the ease of access via an intuitive platform [72] that is directly connected to the financial markets and through which new investment opportunities can be discovered. We find that the most investment-minded category of respondents are those with an annual income of over EUR 24,186. This is the only income group where investors outnumber noninvestors. According to Coffi and Babu [73], wealthier clients bear the cost of traditional financial advisors, while lower-income clients take advantage of the digital alternative, robot advisors.

According to Wati et al. [74], empirical evidence shows that the role of financial technology has a positive and significant effect on financial inclusion. These results indicate that fintech can increase financial inclusion. Based on interviews, fintech products that are often used by MSMEs are third-party payment systems and peer-to-peer (P2P) type of payment systems. The large number of MSME entrepreneurs who have used fintech products in their business shows that the use of financial services in the form of savings accounts by MSME entrepreneurs has an impact on increasing financial inclusion. In our model, the use of mobility applications was eliminated during the iterations.

According to Solars and Swacha–Lech [75], the fintech Adoption Index, expressed as a percentage of the digitally active population, reached 64% for 27 countries around the world in 2019. Millennials are the generation characterized by the highest fintech adoption compared to others. An LR model was used to analyze and evaluate the impact of selected determinants of fintech adoption. The millennials most open to innovative fintech services in Poland are young men with high and very high net incomes who are not driven by the low costs of financial services. They appreciate technological novelties, including the possibility of using a smart watch, and when deciding on the choice of a financial institution, they do not care about the direct opinions of their relatives and friends, but take into account opinions they see on social media. This does not seem to be the case in Romania, where smart watch users are not yet a category of importance for fintech service providers.

The immense potential for innovation in downstream applications stemming from the increased speed and reliance and lower latency of 5G mobile technology could greatly increase the resilience of Europe's SMEs, enhance their growth prospects, and support employment. The availability of 5G technology will also support the European Union's

drive towards carbon neutrality under the European Green Deal, as well as the climate action objectives under the Climate Bank Roadmap of the European Investment Bank (EIB).

Our results show that people who already use e-wallets are more likely to open accounts offered by fintech companies. We hypothesize that customers who are accustomed to e-wallets are more willing to try other fintech services. The likelihood of a customer approaching digital solutions for their financial needs should be increased by previous positive experiences with similar tools. The use of virtual cards was not found to have the same statistical significance as electronic wallets, which raises some questions. In Romania, holding virtual cards does not coincide with the willingness to access fintech services on an extended scale. We can hypothesize that consumers do not perceive the risks and benefits of using virtual cards to be equivalent to that of opening a virtual bank account or using an e-wallet. Instead, they perceive electronic wallets as a more useful service that precedes the need of also having a virtual card.

The COVID-19 outbreak, and the quick spread of the virus globally, led people to change some of their practices, such as using alternative means of payment instead of cash. Several experts' advise to minimize the usage of cash lead people to use other methods of payment, such as mobile wallets. According to Alwi and Saleh [76], mobile wallets comprise a technology that needs to be enabled on users' smartphones, enabling users to store money and perform online transactions directly. As the world slowly moves away from using cash and relies on technology for payments, the mobile e-wallet is one of the best tools. With the increasing use of technology and the transition to alternative payment methods, the mobile e-wallet is one of the most widely used tools. However, the mobile e-wallet is a relatively new tool that has only emerged in recent years and has been little explored.

With respect to fintech accounts, we can conclude that previous investments and preference to use e-wallets are factors to determine utilization of fintech accounts. On the other hand, the age, gender, and income of the respondent, which are demographic traits commonly used as customer identifiers are mostly irrelevant. We consider that investments aimed at incorporating financial services into a digital infrastructure, accompanied by a reduction in activities that require face-to-face contacts with the clients, will increase the approval rate of fintech services and capture a larger share of the potential customer base. Of course, the pricing of the products is also very important, and this should be as well taken into consideration in the sale process.

While the nascent field of fintech research is dedicated to topics such as payments, risk management, crowdfunding, and cryptocurrencies, the means of fintech adoption is still the most important topic. We recognize that there may be many covariates that we do not know about which will influence customer preferences for trying fintech offerings. Some may be beyond the control of the bank management, such as whether the bank belongs to a state or to a private equity owner [32]. Understanding fintech adoption factors, joint support of banks and financial stakeholders, and spreading financial literacy are required to overcome the challenge of creating a practical fintech framework [36].

Romanian banks are recommended to include partnerships with fintech companies in their strategy, not only because they need to follow the digital transformation, but also because they can take advantage of this common interest of customers in digital and mobile applications [36]. In addition, banks need to work closely with internet providers to adopt digital banking models [27]. In the eyes of consumers, fintech services are compared to other online shopping services, where the costs associated with using the online shopping alternative (e.g., excessive fees and delivery costs) are a major deterrent for consumers [69]. Cheap internet access is pivotal in the adoption of internet banking, as we have seen in Ramayah's [31] case, the price associated with implementing and adopting internet banking was of little relevance because of governmental efforts to keep the price of internet access.

There might be an impetus for fintech users also to access more crediting instruments. Szopiński [30] discovered that, in Poland, individuals that have a mortgage or a credit card at a bank are more compelled to also use internet banking than those that only have an

open bank account. Consumers expect e-banking to be faster than its physical alternative and to be designed in such a way as to permit fast learning and easy use of the service; they additionally expect it to be secure, accurate, and user-friendly [26]. Szopiński [30] also found that trust is less important than other variables when using internet banking. Vinayek and Jindal [32] observed that customers judge the quality of internet banking services based on customer care, information quality, efficiency, and service performance. Fintech services such as digital banking could remove one of the main barriers—high banking costs and lack of access to a bank branch—to accessing banking services in underdeveloped or developing regions such as Eastern Europe by introducing an affordable and accessible channel through mobile devices [28].

The results of our analysis can help such entities in a two-fold manner: acquire an image of the potential customer base in Romania and recognize the traits that a likely user of fintech and e-banking would exhibit. While e-banking in emerging and still-developing markets could benefit from lower risk due to lower operational efficiency, leading banks are trying to attract as many customers as possible through e-banking and are following the example of US banks that have successfully implemented e-banking strategies [27].

The results of our study indicate a strong correlation between the desire to have access to some kind of electronic wallet, to invest, and to interact with digital financial services. We have provided a good indicator of the potential needs of fintech users and we have determined which features they are most likely to use. This study also shows which features are a customer afterthought; additionally, we have demonstrated that demographic characteristics have a negligible impact on a given customer's decision to choose fintech. The novelty of our research originates from the careful analysis of the digital service consumption habits of university graduates in an Eastern European country. While the employed methodology is not extravagant, the focus on the usage of digital financial apps and tools that can be seen in the questions that were addressed to the respondents makes this paper a worthwhile contribution to the scientific literature in the context of an industry that is increasingly facilitating digital transactions. The concern for a specific sample that has achieved a certain education level is uncommon and paves the way for future inter-group comparisons.

The focus of this article has been not only on the adoption of digital banking apps, but also on how a variety of products—including P2Pp2p payments, e-commerce platforms, electronic wallets, cryptocurrencies and retail digital investments—alongside online banking, are creating shared consumer demand. The need to access at least some of these services is independent of age and gender and is a signal for financial institutions to implement multifunctional digital systems.

We believe that, based on the foundations we have laid in this article, there is still room for improvement, especially when it comes to empirically measuring the widespread attractiveness of fintech services and their market performance. We acknowledge that a small size such as ours could be the major shortcoming of any statistical modeling procedure. Nemes et al. [77] show that small and medium sample sizes lead to an overestimation of the coefficients. The coefficient bias asymptotically approaches zero as the sample size increases, with the break-away point being somewhere in the range of 400–600 observations. Some methods were suggested to mitigate the small sample bias: bootstrapping, bias correction using weighted regression or a jack-knife estimate, and resorting to penalized logistic regression (as suggested by Makalic and Schmidt [78]). A univariate logistic regression using the variables that have been deemed important can help us to identify the optimal sample size for each variable. Alternatively, Hsieh, Blöch, and Larsen [79] provide a simple method to estimate the appropriate sample size that can be scaled for multivariate logistic regressions using a specified significance level and power.

We felt that we took most precautions to avoid biased results, but we refrained from assuming that the relationship between the predictors and the predictions was anything other than a sigmoidal relationship (Ranganathan, Pramesh, and Aggarwal [80]). Further insights into this topic should experiment with other nonlinear assumptions. If this study

looks towards predicting binary outcomes, it can opt to use alternative methods such as linear probability Poisson regression models, as prescribed by Huang [81], or probit regressions (Hoffman [82]). If the focus of a future paper is to predict the attitudes of consumers towards fintech services with more nuance, then we recommend using discriminant analysis (King [55]), as it has been successfully employed many times in analyzing the banking environment while having stricter conditions. For elaborate categorical datasets with small sample sizes, Sigrist [83] believes that even traditional machine learning models with random effect are a viable choice, especially random forest boosting. Future studies can use the effect size measured in this paper, which represents the magnitude of change in the log–odds, to compute an optimal sample size. We also consider the risk of selecting spurious variables or omitting significant variables in our final model; however, we believe that the piecewise technique that we employed was appropriate. We encourage future studies to work with similar variables and develop more comprehensive questionnaires. Furthermore, user data provided directly by banks or companies working in the field of digital finance or that rely heavily on the engagement of users with digital apps could supplant the need for a qualitative questionnaire. Reliance on hard and objective data can help in more sincerely forecasting the likelihood that customers will convert to a new service. Understanding that basic financial needs such as holding currency in a digital wallet or having access to investment instruments and opportunities drive the adoption of digital financial services may help enterprises focus their future business models foremost on these areas in order to attract customers.

This study, "Fintech Adoption Factors: A Study on an Educated Romanian Population", examines the variables that impact the adoption of fintech services among Romanian individuals with a higher education background. To shed light on the current customer perspective of digitalization in the Romanian banking sector, this study uses quantitative research methodology that includes logistic regression analysis to analyze the extent of mobile banking usage, payments, and banking product needs among the target demographic. The study incorporates findings from a variety of international studies—such as those that look at the US banking system, China's internet banking adoption, and the effect of digital payment methods on financial vulnerability—in order to investigate the elements that influence acceptance and use of fintech. Through the synthesis of these results, this study provides a strong basis for comprehending the worldwide trends in the adoption of fintech, enabling an in-depth examination of the particular case study that is the Romanian population.

## 6. Conclusions

Our research sheds new light on Romania's university graduates' preference for digital financial services. We use data collected from a survey which is further analyzed using a logistic regression model that reveals the links between the demographic features of the population and its previous use and willingness to use electronic devices and digital applications, on the one hand, and the use of financial products, including fintech-based products, on the other hand.

The results obtained from the survey are important for banks to better understand the needs of clients and preferences and to better model their products to obtain market share. The significant interest in online features of products and accessibility for educated people would translate in the need to invest more in technological developments so that customers can use more online products. At the same time, banks can reduce their footprint even more and maintain physical branches or call centers in cities or areas with older people and people with a lower level of education. Banks should calibrate their products to attract savings from highly educated people through online presence and reduce their physical presence for such customers. We notice there is a good amount of trust among highly educated people with a good income in the financial system, enabling further developments.

The respondents for this study were selected from the high-income category; hence, preferences among low-income populations are not represented in this research. This may

be further analyzed for lower-income categories and people with a lower level of education, to understand which products are of higher interest and whether there is interest for online accessibility in these mentioned categories.

Fintech accounts are new in Romania, but the utilization in our sample size is 72%—this is high. There is also interest in opening potential new accounts (70% of the respondents are positive), while for cryptocurrencies the interest is very low still (24%).

Using logistic regression, we discovered that previous financial investments and electronic wallets are factors that influence peoples' decisions to use fintech accounts. We used more variables in our model, such as gender, age, income, previous use of smartwatches, previous use of mobility applications, trust in 5G technologies with regard to financial developments, and previous use of virtual cards; however, none of them were significant in our model with respect to the decision to adopt fintech accounts. Previous studies which were mentioned in this paper suggest that continuous technological developments create favorable conditions for fintech adoption; here, age is a factor that has been found to be negatively correlated with fintech use, while income has been found to be positively correlated. Previous investments were also seen in other studies as factors that influence the decision to use fintech accounts. Previous logistic regression studies have discovered that, in Poland, young men with high and very high incomes and smart watches users have the highest odds for using fintech accounts. In Romania, based on our study, this situation was not proven to be the case.

The novelty of our study is that it was performed for the first time in Romania and only on a sample of the population with a tertiary education level. Even in this distribution, there is still room for more fintech adoption. As the results show that age and income are not factors to influence the decision to adopt fintech accounts, we believe that fintech companies can attract any type of clients and further enlarge their business, they just need improved sales strategies. The same can happen with local banks if they decide to expand their digital services.

With regard to cryptocurrency use, we notice that our sample population has the highest interest in the 29–43- and 44–58-years-old age categories. It is evident that young people do not have the money or experience to invest, and the older generation is not interested due to the high risk it has.

The majority of respondents on this study reported that they do not use cryptocurrencies; therefore, we consider that having a cryptocurrency account cannot influence the use or nonuse of fintech accounts. However, the relation may be the other way around—people that already have fintech accounts may be interested in trading cryptocurrencies. This field is new in economics, and we recommend further study, as cryptocurrencies are an asset class that is worth studying; they are becoming more familiar to the public and trading them through official and regulated platforms may reduce fraud risks.

**Author Contributions:** Conceptualization, L.B., C.A.N. and Z.D.; Methodology, Z.D.; Software, Z.D.; Validation, C.A.N.; Formal analysis, Z.D.; Investigation: C.A.B.; Resources, C.A.B.; Data curation, C.A.N.; Writing—original draft preparation, C.A.B., Z.D. and C.A.B.; Writing—review and editing, C.A.N. and C.A.B.; Visualization, L.B. and D.G.D. All authors have read and agreed to the published version of the manuscript.

**Funding:** This research received no external funding.

**Institutional Review Board Statement:** The proposed study entitled "Fintech adoption factors: a study on an educated Romanian population", co-authored by Lucian Belașcu, Corina Anca Neguț (Cudric), Zeno Dincă, Cosmin Alin Boțoroga, and Dan Gabriel Dumitrescu, has been submitted for ethical evaluation to the Centre of Research in International Business and Economics (CCREI) on 20 June 2022. The ethical commission appointed by the Centre, formed of Professor Rodica Milena Zaharia (president), Professor Anca Gabriela Ilie (member) and Professor Clara Alexandra Volintiru (member) has thoroughly analyzed the study's compliance with ethical considerations and concluded that it meets all ethical standards required for conducting the research. The authors demonstrated that they took into consideration all the aspects related to protecting the anonymity of the responses, free will of participating in the research, and the possibility to withdraw from the research at any

time. Therefore, the questionnaire that will be used in this study to collect data is approved by the Centre before being sent to potential respondents.

**Informed Consent Statement:** Informed consent was obtained from all subjects involved in the study.

**Data Availability Statement:** Data is available within the article and Appendix D.

**Conflicts of Interest:** The authors declare no conflict of interest.

## Appendix A

Questionnaire:

1. What is your gender: male (1)/female (2)
2. What is your age: 18–28 (1); 29–43 (2); 44–58 (3); >59 (4)
3. What are your last studies: post university (doctoral, masters) (1); university (2)
4. In what area do you live: urban (1); rural (2)
5. What is your professional status: employee (1); I prefer not to answer (2); entrepreneur (3); freelancer (4);
6. What is your annual income: < 30,000 (1); 30,000–53,999 (2); 54,000–89,999 (3); 90,000–118,999 (4); >119,000 RON (5)
7.1 Do you or your family use Smart TV? No (1)/Yes (2)
7.2 Do you or your family use Smartphone? No (1)/Yes (2)
7.3 Do you or your family use TV applications? No (1)/Yes (2)
7.4 Do you or your family use mobility applications? No (1)/Yes (2)
7.5 Do you or your family use online banking? No (1)/Yes (2)
7.6 Do you or your family use Smartwatch? No (1)/Yes (2)
7.7 Do you or your family use Cryptocurrencies? No (1)/Yes (2)
8.1 What type of banking products do you use: current account? No (1)/Yes (2)
8.2 What type of banking products do you use: card payment at merchants? No (1)/Yes (2)
8.3 What type of banking products do you use: credit cards? No (1)/Yes (2)
8.4 What type of banking products do you use: payments through internet/mobile banking? No (1)/Yes (2)
8.5 What type of banking products do you use: personal needs, auto loans, leasing? No (1)/Yes (2)
8.6 Other financial services (investment in state bonds, mutual funds) No (1)/Yes (2)
9. If you had savings above 10,000 EUR, how would you use them? Financial investments (state bonds, mutual funds, stocks) (1); Opening saving accounts/bank deposits (2); buy a car (3); investments in my personal development: travels, sabbatical year, courses (4); savings at home (5); refurbishment of my house (6); I don't know (7)
10. Do you trust banks authorized by NBR? not at all (1); very few (2); a few (3); a lot (4); very much (5)
11. Do you trust 5G technology to help the development of financial services? not at all (1); very few (2); a few (3); a lot (4); very much (5)
12. With how many banks do you work now? (1; 2; 3; 4 or more)
13. What are the criteria to use when choosing the bank you work with? (1) to have branches close to home/office (2); friends or family recommendation (4); possibility to open online accounts or online access of services (1); top 10 placement (assets, image, trust) (3); interest paid/received (5); none (6)
14. Do you use online/mobile banking services? No (1)/Yes (2)
15. During time, using Online/Mobile banking application, did you discover and utilized new services that you did not use before? Yes (1)/No (2)/I don't know (3)
16. What instrument do you use to access Online/Mobile banking application: smartphones (1), laptop (2), desk PC (3), tablet (4)
17. In the following 12 months do you intend to take a credit card? For sure (1); maybe yes (2); I don't know (3); maybe no (4); for sure not (5)
18. In the following 12 months do you intend to take a personal needs/real estate loan? For sure (1); maybe yes (2); I don't know (3); maybe no (4); for sure not (5)
19. How do you prefer to open a bank account? I prefer online (1); I prefer direct interaction (2); I am indifferent (3); I don't know (4)
20. How do you prefer to open a bank deposit/loan? I prefer online (1); I prefer direct interaction (2); I am indifferent (3); I don't know (4)
21. How do you prefer to perform payment of taxes, invoices, bills, buying of tickets I prefer online (1); I prefer direct interaction (2); I am indifferent (3);
22. How often do you use the direct interaction with bank officer? Never or almost never (1); once a year (2); few times a year (3); once a month (4); daily or weekly (5)
23. How often do you use customer phone relationship with bank officer? Never or almost never (1); once a year (2); few times a year (3); once a month (4);
24. How often do you use the mobile banking application (smartphone or tablet) to interact with your bank? Never or almost never (1); once a year (2); few times a year (3); once a month (4); daily or weekly (5)
25. How often do you use the internet banking application (desk PC or laptop) to interact with your bank? Never or almost never (1); once a year (2); few times a year (3); once a month (4); daily or weekly (5)
26. With regards to your card, how important is it for you to use it in the electronic wallet? I am not interested (1); indifferent (2); important (3); very important (4);

27. With regards to your card, how important is it for you to withdraw money from ATM without physical card? I am not interested (1); indifferent (2); important (3); very important (4);
28. With regards to your card, how important is it for you to issue a virtual card in your online/mobile banking application? I am not interested (1); indifferent (2); important (3); very important (4);
29. How important are for you the actions your bank performs concerning environment/social life? I am not interested (1); indifferent (2); important (3); very important (4);
30. Do you use financial services by Fintech companies (other than the ones authorized by Romanian Central Bank) (Orange money, Monese, Revolut, Paysera) No (1)/Yes (2)
31. If you answered yes before, what are the services you use: card payment abroad (2); fast payments to friends (1); FX transactions at better rates (3); I receive income from abroad (5); others (4)
32. Would you be interested in working with Fintech companies? Totally disagree (1); partially disagree (2); I don't know (3); partially agree (4); totally agree (5);
33. Would you be interested in working with entities that offer decentralized services (blockchain technology, less compliance and less regulations)? No (1); I would be skeptical (2); Yes (3)
34. Concerning daily services you are using (hairdressers, house cleaning, massage, children daycare, some restaurants) how satisfied are you concerning the impossibility to pay by card? Very unsatisfied (1); unsatisfied (2); indifferent (3); I am not interested, I can pay by cash (4)

## Appendix B

**Table A1.** Questionnaire statistics.

| Variable | | Descriptive Statistics | | | | | | | |
|---|---|---|---|---|---|---|---|---|---|
| | | Valid N | Mean | Median | Mode | Frequency of Mode | Variance | Std.Dev. | Skewness | Kurtosis |
| 1. | Gender | 118 | 1.475 | 1 | 1 | 62 | 0.251 | 0.501 | 0.103 | −2.024 |
| 2. | Age | 118 | 2.339 | 2 | 2 | 62 | 0.517 | 0.719 | 0.233 | −0.068 |
| 3. | Last studies | 118 | 1.322 | 1 | 1 | 80 | 0.220 | 0.469 | 0.772 | −1.429 |
| 4. | Living location | 118 | 1.034 | 1 | 1 | 114 | 0.033 | 0.182 | 5.218 | 25.660 |
| 5. | Professional status | 118 | 1.356 | 1 | 1 | 97 | 0.693 | 0.832 | 2.234 | 3.701 |
| 6. | Annual income | 118 | 3.898 | 5 | 5 | 63 | 2.007 | 1.417 | −0.992 | −0.358 |
| 7.1 | | 118 | 1.797 | 2 | 2 | 94 | 0.163 | 0.404 | −1.493 | 0.232 |
| 7.2 | | 118 | 1.958 | 2 | 2 | 113 | 0.041 | 0.202 | −4.602 | 19.512 |
| 7.3 | | 118 | 1.873 | 2 | 2 | 103 | 0.112 | 0.335 | −2.268 | 3.197 |
| 7.4 | | 118 | 1.864 | 2 | 2 | 102 | 0.118 | 0.344 | −2.156 | 2.695 |
| 7.5 | | 118 | 1.924 | 2 | 2 | 109 | 0.071 | 0.267 | −3.234 | 8.604 |
| 7.6 | | 118 | 1.542 | 2 | 2 | 64 | 0.250 | 0.500 | −0.172 | −2.005 |
| 7.7 | | 118 | 1.144 | 1 | 1 | 101 | 0.124 | 0.353 | 2.053 | 2.254 |
| 8.1 | | 118 | 1.966 | 2 | 2 | 114 | 0.033 | 0.182 | −5.218 | 25.660 |
| 8.2 | | 118 | 1.941 | 2 | 2 | 111 | 0.056 | 0.237 | −3.779 | 12.494 |
| 8.3 | | 118 | 1.653 | 2 | 2 | 77 | 0.229 | 0.478 | −0.649 | −1.606 |
| 8.4 | | 118 | 1.907 | 2 | 2 | 107 | 0.085 | 0.292 | −2.834 | 6.137 |
| 8.5 | | 118 | 1.424 | 1 | 1 | 68 | 0.246 | 0.496 | 0.313 | −1.935 |
| 8.6 | | 118 | 1.432 | 1 | 1 | 67 | 0.248 | 0.497 | 0.277 | −1.957 |
| 9. | Savings | 118 | 2.186 | 2 | 1 | 57 | 2.580 | 1.606 | 1.427 | 1.169 |
| 10. | Trust in banks | 118 | 4.305 | 5 | 5 | 63 | 0.863 | 0.929 | −1.493 | 2.072 |
| 11. | Trust in 5G | 118 | 4.034 | 5 | 5 | 59 | 1.127 | 1.062 | −0.504 | −1.063 |
| 12. | How many banks | 118 | 2.479 | 2 | 2 | 43 | 0.924 | 0.961 | 0.091 | −0.923 |
| 13. | Bank choosing criteria | 118 | 1.737 | 1 | 1 | 65 | 0.965 | 0.982 | 1.157 | 0.192 |
| 14. | Do you use OB/MB? | 118 | 1.958 | 2 | 2 | 113 | 0.041 | 0.202 | −4.602 | 19.512 |
| 15. | Did you discover new features? | 118 | 1.297 | 1 | 1 | 90 | 0.330 | 0.575 | 1.814 | 2.270 |
| 16. | Instrument used for OB/MB | 118 | 3.619 | 3 | 1 | 58 | 7.811 | 2.795 | 0.387 | −1.280 |
| 17. | Credit card plans | 118 | 2.398 | 2 | 1 | 55 | 2.037 | 1.427 | 0.236 | −1.678 |
| 18. | Loan plans | 118 | 1.992 | 2 | 1 | 48 | 1.171 | 1.082 | 1.005 | 0.215 |
| 19. | Bank account opening | 118 | 1.415 | 1 | 1 | 85 | 0.535 | 0.732 | 1.564 | 1.238 |
| 20. | Deposit/Loan opening | 118 | 1.458 | 1 | 1 | 78 | 0.507 | 0.712 | 1.381 | 0.966 |
| 21. | Tax payments | 118 | 1.110 | 1 | 1 | 108 | 0.150 | 0.387 | 3.741 | 13.949 |
| 22. | Direct bank interactions | 118 | 2.542 | 3 | 3 | 49 | 1.430 | 1.196 | −0.178 | −1.298 |
| 23. | Call center | 118 | 1.898 | 2 | 2 | 56 | 0.878 | 0.937 | 1.093 | 0.509 |
| 24. | MB usage | 118 | 4.393 | 5 | 5 | 87 | 1.534 | 1.238 | −2.036 | 2.809 |

**Table A1.** *Cont.*

| Variable | | Descriptive Statistics | | | | | | | | |
|---|---|---|---|---|---|---|---|---|---|---|
| | | Valid N | Mean | Median | Mode | Frequency of Mode | Variance | Std.Dev. | Skewness | Kurtosis |
| 25. | OB usage | 118 | 3.254 | 3 | 5 | 33 | 2.277 | 1.509 | −0.382 | −1.225 |
| 26. | Electronic wallet | 118 | 4.059 | 5 | 5 | 59 | 1.270 | 1.127 | −0.847 | −0.607 |
| 27. | Cardless transactions | 118 | 2.559 | 3 | 2 | 39 | 0.949 | 0.974 | −0.029 | −0.975 |
| 28. | Virtual cards | 118 | 2.754 | 3 | 3 | 50 | 0.854 | 0.924 | −0.349 | −0.667 |
| 29. | Bank ESG actions | 118 | 2.703 | 3 | 3 | 60 | 0.689 | 0.830 | −0.398 | −0.265 |
| 30. | Fintech usage | 118 | 1.619 | 2 | 2 | 73 | 0.238 | 0.488 | −0.495 | −1.786 |
| 31. | Fintech products | 74 | 1.919 | 2 | 2 | 51 | 0.459 | 0.678 | 1.457 | 5.842 |
| 32. | Would you work with Fintech's | 118 | 3.737 | 4 | 4 | 55 | 1.221 | 1.105 | −1.007 | 0.537 |
| 33. | Would you work with crypto? | 118 | 1.932 | 2 | 2 | 54 | 0.542 | 0.736 | 0.108 | −1.134 |
| 34. | Cash preference | 118 | 1.814 | 2 | 1 | 53 | 0.820 | 0.905 | 0.942 | 0.080 |

## Appendix C

**Table A2.** LR model variables descriptive statistics.

| | N | Mean | Std. Deviation | Variance | Skewness | Kurtosis |
|---|---|---|---|---|---|---|
| | Statistic | Statistic | Statistic | Statistic | Statistic | Statistic |
| Gen | 118 | 1.4746 | 0.50148 | 0.251 | 0.103 | −2.024 |
| Age | 118 | 2.3390 | 0.71874 | 0.517 | 0.233 | −0.068 |
| Inc | 118 | 3.8983 | 1.41657 | 2.007 | −0.992 | −0.358 |
| MobApp | 118 | 1.8644 | 0.34382 | 0.118 | −2.156 | 2.695 |
| SmartW | 118 | 1.5424 | 0.50033 | 0.250 | −0.172 | −2.005 |
| Inv | 118 | 1.4322 | 0.49749 | 0.248 | 0.277 | −1.957 |
| fintech | 118 | 1.6186 | 0.48779 | 0.238 | −0.495 | −1.786 |
| Tec5G | 118 | 4.0339 | 1.06162 | 1.127 | −0.504 | −1.063 |
| EWallet | 118 | 4.0593 | 1.12692 | 1.270 | −0.847 | −0.607 |
| VCard | 118 | 2.7542 | 0.92391 | 0.854 | −0.349 | −0.667 |

## Appendix D

Data is available at the following link:

https://docs.google.com/spreadsheets/d/1aRs1GJfdk_ZkC1m9vFHEC5UmDDz1rgXU/edit#gid=1809903748.

Transformed data for the logit analysis is at the following link:

https://docs.google.com/spreadsheets/d/177JfB-37wXt45hzlMcIVwty73kkeqDg_/edit#gid=1859382997.

**Table A3.** Coefficient variance–decomposition matrix.

| Dimension | Eigenvalue | Condition Index | Variance Proportions | | | | | | | | | |
|---|---|---|---|---|---|---|---|---|---|---|---|---|
| | | | (Constant) | Gen | Age | Inc | MobApp | SmartW | Inv | Tec5G | EWallet | VCard |
| 1 | 9.331 | 1.000 | 0.00 | 0.00 | 0.00 | 0.00 | 0.00 | 0.00 | 0.00 | 0.00 | 0.00 | 0.00 |
| 2 | 0.172 | 7.358 | 0.00 | 0.19 | 0.05 | 0.12 | 0.00 | 0.00 | 0.08 | 0.00 | 0.00 | 0.00 |
| 3 | 0.144 | 8.036 | 0.00 | 0.01 | 0.13 | 0.01 | 0.00 | 0.06 | 0.06 | 0.03 | 0.03 | 0.12 |
| 4 | 0.094 | 9.979 | 0.00 | 0.27 | 0.33 | 0.01 | 0.00 | 0.00 | 0.04 | 0.02 | 0.02 | 0.08 |
| 5 | 0.075 | 11.143 | 0.00 | 0.02 | 0.05 | 0.03 | 0.00 | 0.39 | 0.00 | 0.30 | 0.02 | 0.07 |
| 6 | 0.065 | 11.941 | 0.00 | 0.10 | 0.01 | 0.59 | 0.00 | 0.24 | 0.17 | 0.07 | 0.00 | 0.01 |
| 7 | 0.061 | 12.396 | 0.00 | 0.08 | 0.00 | 0.11 | 0.00 | 0.21 | 0.42 | 0.26 | 0.00 | 0.08 |
| 8 | 0.031 | 17.337 | 0.00 | 0.02 | 0.00 | 0.03 | 0.00 | 0.03 | 0.00 | 0.04 | 0.89 | 0.52 |
| 9 | 0.019 | 22.405 | 0.01 | 0.15 | 0.01 | 0.08 | 0.85 | 0.04 | 0.17 | 0.05 | 0.03 | 0.11 |
| 10 | 0.008 | 34.327 | 0.99 | 0.16 | 0.42 | 0.00 | 0.14 | 0.02 | 0.03 | 0.22 | 0.01 | 0.01 |

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
