# Peer review of "Fintech Adoption Factors: A Study on an Educated Romanian Population"

_societies, doi:10.3390/soc13120262_

Round 1

Reviewer 1 Report

Comments and Suggestions for Authors

The literary background needs to be supplemented with digital services and not only banking. Perhaps it would be appropriate to include the DESI indicator, one part of which is also devoted to electronic banking. A part of the text is needed from the introduction. which refers to the methodology, move it to the methodology and modify the introduction with an introduction to the issue and justification of the article. In the methodology, state why only respondents with tertiary education took part in the questionnaire survey.

Apparently, there was an error with the graphs, how can an 18-year-old person have a completed university education? On page 13, it is stated that at most 53% of the respondents fall in the age category

28-43 years old, but such a category is not found in the chart.

It is necessary to revise the result part so that it corresponds to the comments in the text.

The article deals with an interesting topic, but it is necessary to revise the result part of the contribution and supplement the literature review with European comparisons of other countries, not only the comparison with the USA.

Comments on the Quality of English Language

There are minor stylistic errors in the text.

Reviewer 2 Report

Comments and Suggestions for Authors

The paper presents the results of a study examining the relationship between the choice to use Financial Technology (Fintech) and the level of education among the population in Romania. The data collected is derived from a survey conducted among the participants, and the analysis is based on a logistic regression model, revealing the associations between the demographic characteristics of the population and their prior use and willingness to use electronic devices and digital applications, as well as their utilization of financial products, including those based on Fintech technology.

During the review of the text, several comments and suggestions should be taken into account:

  • Introduction: The text lacks a clear introduction that would define what Fintech is and why it is an important research area. The introduction can also include the research's objectives and research questions, helping readers better understand the context of the work.
  • Lack of Reference to Theoretical Background: There is no reference to the theoretical background in the Fintech field. Presenting existing theories, concepts, and research in the area can provide readers with context and help them understand why the study is relevant.
  • Further Development of Results: In the "Conclusions" section, the main research findings are discussed, but further analysis and discussion of why these results are significant and what implications they may have for the banking industry and the Fintech market are missing.
  • Bibliography: The bibliography is missing, as well as references to sources of information and research used during the creation of the work. Including a bibliography will allow readers to verify sources and learn more about existing research in the Fintech field.
  • Editing and Consistency: Checking the correctness of grammar, punctuation, and style throughout the text can make it more readable and professional.
  • Text Structure: The paper could benefit from a clearer structure, including sections such as Introduction, Theoretical Background, Methodology, Results, Discussion, and Conclusions. This will help readers better understand the flow of the argument and analysis.
  • Additional Analysis: Consider whether additional analyses can be conducted to better understand the relationships between variables and factors influencing Fintech adoption. For example, it may be possible to investigate why interest in cryptocurrencies is relatively low in the studied population.
  • International Context: The study focuses on the situation in Romania, but adding an international context or comparing the results with other countries could provide interesting perspectives.
  • Development of Argumentation: In some parts of the text, it would be valuable to develop the argumentation and provide more detailed explanations, especially in the "Conclusions" section.
  • Summary: The text can benefit from a clear summary that highlights the main conclusions and key points that the reader should remember.

Overall, the paper contains valuable information about Fintech adoption in Romania among the educated population, but it requires some improvements to become more reader-friendly and accessible.

Author Response

please see attachement; we have mixed the answers as some of them are similar

Round 2

Reviewer 1 Report

Comments and Suggestions for Authors

The article is fine after editing, but it has a formal error. Key words are missing in the abstract.

Author Response

Hi,

Please see attached our response for both comments.

Reviewer 2 Report

Comments and Suggestions for Authors

i am satisfied with changes 

Author Response

Hi,

Please see attached our response to both comments.

Thank you
